# Prognostic Features of Sarcopenia in Older Hospitalized Patients: A 6-Month Follow-Up Study

**DOI:** 10.3390/jcm13113116

**Published:** 2024-05-26

**Authors:** Anne Ferring, Luisa Mück, Jill Stegemann, Laura Wiebe, Ingrid Becker, Thomas Benzing, Anna Maria Meyer, Maria Cristina Polidori

**Affiliations:** 1Department II of Internal Medicine and Center for Molecular Medicine Cologne, Faculty of Medicine and University Hospital Cologne, University of Cologne, 50937 Cologne, Germany; anneferring@hotmail.de (A.F.);; 2Institute of Medical Statistics and Computational Biology, University Hospital of Cologne, 50937 Cologne, Germany; 3CECAD, Faculty of Medicine and University Hospital Cologne, University of Cologne, 50937 Cologne, Germany

**Keywords:** sarcopenia, EWGSOP2, MPI, prognosis, frailty, comprehensive geriatric assessment, older people

## Abstract

**Background:** Sarcopenia is associated with adverse health outcomes. Understanding the association between sarcopenia, multidimensional frailty, and prognosis is essential for improving patient care. The aim of this study was to assess the prevalence and prognostic signature of sarcopenia in an acute hospital setting co-led by internists and geriatricians. **Methods:** Sarcopenia was assessed by applying the European Working Group on Sarcopenia in Older People (EWGSOP2) algorithm, including the SARC-F score, handgrip strength, bioelectrical impedance analysis (BIA), and Timed Up and Go (TUG) test, to 97 older multimorbid inpatients (76.5 ± 6.8 years, 55% women). The patients underwent a Comprehensive Geriatric Assessment (CGA) including an evaluation of Geriatric Syndromes (GSs) and Resources (GR) and prognosis calculation using the CGA-based Multidimensional Prognostic Index (MPI), European Quality of life—5 Dimensions (EQ-5D-5L) scale, Rosenberg Self-Esteem Scale (RSES), and Geriatric Depression Scale (GDS). Information on survival and rehospitalizations was collected 1, 3, and 6 months after discharge. **Results:** Sarcopenia was present in 63% (95% CI: 54–72%) of patients and categorized as probable (31%), confirmed (13%), and severe sarcopenia (18%). Sarcopenic patients showed significantly higher median MPI-values (*p* < 0.001), more GSs (*p* = 0.033), fewer GR (*p* = 0.003), lower EQ-5D-5L scores (*p* < 0.001), and lower RSES scores (*p* = 0.025) than non-sarcopenic patients. Six months after discharge, being sarcopenic at baseline was predictive of falls (*p* = 0.027) and quality of life (*p* = 0.043), independent of age, gender, and MPI. **Conclusions:** Sarcopenia is highly prevalent in older hospitalized multimorbid patients and is associated with poorer prognosis, mood, and quality of life up to 6 months after discharge, independent of age, sex, and MPI as surrogate markers of biological age.

## 1. Introduction

Ongoing demographic change is disclosing an unprecedented clinical landscape, which includes rapidly increasing mortality rates due to multimorbidity [1] and a greater prevalence of geriatric syndromes. Among these changes, the prevalence of sarcopenia, defined as a generalized and progressive, age-related decline in muscle mass, strength, and function [2], is constantly rising. While prevalence estimates might be largely underdiagnosed due to a lack of universally accepted diagnostic algorithms for long periods, solid evidence links sarcopenia to adverse outcomes such as frailty, rehospitalization, mortality, falls, loss of independence, and depression [3,4,5]. Since 2018, the definition and stepwise diagnostic approach of the European Working Group on Sarcopenia in Older People 2 (EWGSOP2) [6] has provided a standardized research approach to the question of the adverse outcomes and prognosis of sarcopenia. The prognosis of older patients, crucial for avoiding frailty and iatrogenic conditions during clinical decision making, is difficult to accurately evaluate due to the increasingly biopsychosocial and heterogeneous nature of poor outcomes with advancing age. Several Comprehensive Geriatric Assessment (CGA)-based risk scores have proven more useful than illness-centered tools for predicting disease trajectories in older adults [7,8,9]. Among these, the Multidimensional Prognostic Index (MPI) [10] is a well-established, highly validated tool used to quantify patients’ multidimensional frailty and prognosis [7,11]. The aim of the present investigation was to evaluate the prevalence and prognostic blueprint of sarcopenia as defined by the EWGSOP2 criteria in older multimorbid patients undergoing geriatric treatment in a recently established acute hospital setting co-led by internists and geriatricians. 

## 2. Methods

Study Design and Patients: To achieve the above-described aim, data obtained from older multimorbid inpatients included in an RCT investigating the effects of a tailored individualized discharge program (TIDP) compared with usual rehabilitative care [12] in an acute clinical ward co-led by geriatricians and internists were used. As a recently established innovative ward of the large metropolitan University Hospital of Cologne designed to cover the needs of the older population of multimorbid patients requiring potentially disabling disease-centered high-performance medicine [12], patients undergo early complex geriatric rehabilitation (ECGR), and the ward is co-led by internists and geriatricians equally responsible for clinical decision making [12]. Patients 60 years old or older were included if they were admitted due to acute disease or relapse of a chronic condition, multimorbid (with synchronous presence of two or more chronic conditions that require long-term treatment [13]), and suffering from at least two geriatric syndromes requiring typical rehabilitative care. 

For the purpose of the present analysis, patients with incomplete SARC-F-scores (n = 1), without values for handgrip strength (n = 4), and who had not undergone Bioelectrical Impedance Analysis (BIA, n = 8) were excluded from the secondary analysis. The final sample size comprised 97 patients (Figure 1).

Study Procedures: As previously described, patients underwent, together with acute disease standard-of-care treatment and ECGR, a CGA that included an assessment of the MPI [10], calculated based on its eight subdomains: the Activities of Daily Living (ADL) and Instrumental Activities of Daily Living (IADL) for the functional status, the Cumulative Illness Rating Scale (CIRS) for comorbidities, Exton Smith Scale (ESS) for pressure score risk, Mini Nutritional Assessment Short Form (MNA-SF) for nutritional status, the Short Portable Mental Status Questionnaire (SPMSQ) for cognitive status, and medication intake and living condition. By summing the values of each respective severity domain and dividing the sum by 8, the MPI value was obtained, and patients were allocated into three risk groups. These groups indicate the risk of mortality and other adverse outcomes, including hospitalization, admission to long-term care facilities, or homecare need. MPI-1 patients have a score between 0 and 0.33, indicating a low risk; MPI-2 patients have a score of 0.34–0.66, indicating moderate risk; and MPI-3 patients have a score of 0.67–1, indicating they are at high risk [10,11,14]. Additionally, Geriatric Syndromes (GS) and Resources (GR) [15] and Patient-Related-Outcome-Measures (PROMs) including Geriatric Depression Scale (GDS) [16], European Quality of life-5 Dimensions (EQ-5D-5L) for health-related quality of life (HRQoL) [17], and Rosenberg Self-Esteem Scale (RSES) [18] scores were collected for all patients. For the assessment of quality of life, the German version of the EQ-5D-5L, which generates an index score on a scale from −0.66 (representing a state worse than death) to 1 (indicating full health), was used [17]. The RSES is a self-assessment that is evaluated through a set of ten statements, resulting in a score between 0 and 30, while scores below 15 indicate low self-esteem [18]. Information about hospitalizations and falls during the year preceding admission, nursing needs (grade of care according to the German nursing care insurance, spanning from grade 0 to 5 [19]), and number of ECGR days was collected. Information on laboratory values including albumin and total protein levels on admission and discharge was also available. Patients were assigned randomly to either receive ECGR alone or in combination with a tailored intersectional discharge program (TIDP). The TIDP group received an interprofessional discharge treatment plan, which included personalized counselling tailored to various aspects and a reference guidebook. One, three, and six months after discharge, all patients underwent follow-up via telephone to collect information about mortality, rehospitalizations, falls, and nursing needs, as well as EQ-5D-5L and GDS.

Diagnosis of Sarcopenia: The assessment of sarcopenia was performed using the EWGSOP2 algorithm [6] (Figure 2). In Step A, the patients’ scores on the SARC-F scale, consisting of five items, namely, strength, falls, walking assistance, rise from a chair, and stair climbing, were evaluated. With a total score ranging from 0 to 10 points, patients with ≥4 points are expected to be at higher risk for sarcopenia [20] and were included in Step B. In Step B, measurement of muscle strength was assessed via handgrip strength in kg [21] for both sides using a standardized Jamar dynamometer. Patients performed maximal isometric contractions three times with both hands, with 15 s breaks in between. The highest value was used as the handgrip strength value in the present analysis. If handgrip strength was low (<16 kg for women and <27 kg for men), the corresponding patient’s status was classified as Probable Sarcopenia (PS) and they were included in Step C. In Step C, BIA-determined body composition and muscle mass were analyzed, using the BIACORPUS RX 4004M developed by MEDI CAL HealthCare GmbH at the time of admission. For a valid estimation of Appendicular Skeletal Muscle Mass (ASMM), the equation created by Sergie et al. was used: ASMM (kg) = −3.964 + (0.227 × (height^2^/Resistance)) + (0.095 × weight) + (1.384 × sex (with m = 1; f = 0)) + (0.064 × Reactance) [22]. ASMM was normalized in relation to each subject’s height to obtain the ASMM index in kg/m^2^. After the EWGSOP2 cut-off points for low muscle quantities (<5.5 kg/m^2^ for women and <7 kg/m^2^ for men [6]) were applied, patients were diagnosed with Confirmed Sarcopenia (CS) and were included in Step D. In Step D, TUG test was used for quantifying severity of sarcopenia [23]: the time taken by a participant to rise from a sitting position, walk 3 m, return, and sit down again was measured with a stopwatch in seconds. As cut-off points for both women and men, ≥20 s indicated Severe Sarcopenia (SS). 

Statistical Analysis: All statistical analyses were performed using IBM SPSS (Statistical Package for Social Sciences, SPSS Inc., Chicago, IL, USA, version 28.0, Property of IBM Corp.) software. Descriptive statistics are expressed in absolute numbers and relative frequencies for categorical variables. Mean (standard deviation (SD)) and median (interquartile range (IQR)) are used for numerical variables. The Kolmogorov–Smirnov test was performed to test for normal distribution. Univariate tests such as Chi-square and Fisher’s exact test were used to test for associations of frequencies, using univariate ANOVA for means and Mann–Whitney-U for medians.

Linear and logistic regression analyses were used to test the influence of sarcopenia groups at admission on outcomes and were adjusted for age, sex, and MPI on admission, unless otherwise specified. FU outcomes were further adjusted for intervention (TIDP) and the respective scales on admission whenever assessed. Bonferroni-corrected post hoc analysis, a univariate test, was used for 4-group comparison between the different groups for sarcopenia. Significance level was set at 0.05 (two-sided) to account for multiple pairwise comparisons.

Registration, Participant Consent, and Ethics: The study that this secondary analysis is based upon is registered at the German Clinical Trials Register (DRKS00015996), and the authors declare that this study respects the ethical standards for human experimentation that are stated in the Declaration of Helsinki of 1975, as revised in 2000, as well as the national law. The original RCT was approved by the Ethical Committee (EK 18-394) of the University Hospital of Cologne, Germany, and each patient or proxy respondent signed informed consent forms. 

## 3. Results

### 3.1. Characteristics of the Study Population

The demographic and clinical characteristics of the 97 patients are presented in Table 1. Mean age of the study population was 76.5 (SD 6.8) years, 55% were female. In Step A and B, 37% (95% CI: 28–48%) of the patients (n = 36) were diagnosed as having no sarcopenia. In Step B, 31% (95% CI: 23–40%) (n = 30) had SARC-F scores ≥4 and low handgrip and were diagnosed as Probable Sarcopenia (PS). As 13% (95% CI: 7–21%) (n = 13) additionally showed low ASMM in the BIA, they were diagnosed as Confirmed Sarcopenia (CS) in Step C. In Step D, according to TUG test, 19% (95% CI: 11–26%) of the patients (n = 18) were diagnosed with Severe Sarcopenia (SS). Prevalence of sarcopenia in the study sample therefore was 63% (95% CI: 54–72%), including PS, CS and SS (Figure 2). 

### 3.2. Association between Sarcopenia, Prognosis, and Outcomes

The median MPI value upon admission was 0.56, indicating moderate frailty and risk of poor outcomes at follow up, with sarcopenic patients showing significantly higher MPI values, independently of age and sex, compared to non-sarcopenic patients (*p* < 0.001, Table 1, Figure 3). Linear regression analysis revealed a significant difference between MPI values upon admission for the non-sarcopenic group and SS patients (*p* < 0.001) as well as between PS and SS patients (*p* = 0.009). With respect to the MPI subdomains upon admission and adjusted for age and sex, sarcopenic patients presented significantly higher CIRS values (6.7 vs. 5.7, *p* = 0.011), lower ADL (3.2 vs. 5.1, *p* < 0.001) and IADL (3.1 vs. 5.9, *p* < 0.001) values, lower ESS (14.4 vs. 16.8, *p* = 0.003) values, lower MNA-SF (6.8 vs. 8.2, *p* = 0.024) values, and higher SPMSQ (2.1 vs. 0.9, *p* = 0.011) values compared to non-sarcopenic patients. Bonferroni-corrected post hoc tests revealed a significant difference between the ADL values of non-sarcopenic and PS (*p* < 0.001, M_Diff_ = 1.72, 95%-CI[0.73–2.71]), CS (*p* = 0.018, M_Diff_ = 1.47, 95%-CI[0.17–2.76]), and SS patients (*p* < 0.001, M_Diff_ = 2.42, 95%-CI[1.32–3.63]), as well as significant differences in the remaining MPI subdomains (Table 2). All the *p*-values of significant differences of the pairwise comparison between the groups are shown in Table 2. RSES (*p* = 0.025) and EQ-5D-5L (*p* < 0.001) values upon admission were significantly lower in patients with sarcopenia compared to those for non-sarcopenic patients. Patients with sarcopenia presented significantly more GSs (*p* = 0.033) and fewer GR (*p* = 0.003) and a higher prevalence of grade of care at admission (*p* = 0.012) and needed more ECGR days (18.5 vs. 13.7 days, *p* = 0.006) when compared to non-sarcopenic patients (Table 1). Patients with sarcopenia had significantly lower serum albumin levels at the time of admission (*p* < 0.001, Table 1) than non-sarcopenic patients. Patients with sarcopenia suffered significantly more falls in the preceding year (*p* = 0.008) and during FUs (*p* = 0.004) independently of age, gender, MPI upon admission and discharge, and TIDP. Also, patients with sarcopenia had lower EQ-5D-5L scores (*p* < 0.001) in all follow-ups; after one month, higher GDS scores (*p* = 0.035); and, after three months, lower RSES scores (*p* = 0.002) when compared to non-sarcopenic patients (Table 1) and after adjustment for age, gender, MPI, and TIDP.

## 4. Discussion

The present study shows a high prevalence of sarcopenia within an over-60 population segment admitted to an acute geriatric ward. Furthermore, this analysis highlights the clinical relevance of sarcopenia in terms of PROMs, quantified prognosis, and multidimensional frailty. A longitudinal association between sarcopenia’s severity and life-determining outcomes, such as further falls, but also quality of life and self-esteem, was shown. 

To the best of our knowledge, this is the first report of a significant association between sarcopenia, diagnosed according to the EWGSOP2-algorithm; multidimensional prognosis and frailty measured using the MPI; and health outcomes at three different timepoints. The MPI was shown to be highly accurate in forecasting poor outcomes with advancing age in different patient populations [7,11,24,25]. For the present group of patients, the MPI was shown to be able to accurately monitor the success of ECGR vs. usual care administered to similarly frail patient populations [12]. Importantly, this study showed that the longitudinal changes quantified by the MPI are clinically relevant in terms of rehospitalization and mortality rates up to 6 months after discharge [12]. The ability of MPIs to capture clinically relevant outcomes in the longitudinal health trajectory of older patients with sarcopenia is also clear in this analysis. In fact, one further major observation of the present investigation is the dose–response association between MPI scores and sarcopenia severity and related outcomes at follow-ups. This result is not surprising from a pathophysiological point of view, as a highly sensitive multidimensional tool can seize at best the multifactorial, individual, dynamic nature of the aging process [11], especially under stress conditions such as those in hospitalization. However, there are few clinical studies using a quantifiable, scaled CGA that can be used as a readout to address the clinical impact of sarcopenia in terms of multidimensional frailty and prognosis severity. 

Most of the MPI subcategories were shown to be closely interwoven with sarcopenia. Among these, the role of nutrition is particularly noteworthy. There is substantial evidence demonstrating a strong correlation between nutrition and muscle mass, strength, and overall function in older adults, underscoring the critical significance of nutrition both in preventing and managing sarcopenia [26]. 

As mentioned above and in line with the literature [5,27], one more crucial finding yielded by the present analysis is the longitudinal association of sarcopenia severity not only with previous falls but, importantly, with life-determining outcomes such as further falls in FUs when compared to non-sarcopenic patients. This information is particularly relevant from the perspective of its independence from chronological age, sex, and the MPI. Not only does the MPI—like every CGA—act as a surrogate marker of biological age [7], but it contains several sarcopenia-related factors that are usually difficult to disentangle in real life, such as cognitive performance and nutritional status [7]. The independent nature of sarcopenia as a risk condition for falls underlines the importance of its precise diagnosis at every possible medical visit. 

There are further important observations from this analysis that should be discussed. First of all, an association between sarcopenia and the presence of more GSs [28] was confirmed. With sarcopenia being ICD-10-listed since 2018 in Germany and recognized as an independent disease, its influence on other GSs may become the center of interest of studies in the future. Moreover, it was shown for the first time that significantly more patients with sarcopenia had a grade of care at the time of admission to the hospital. The concept of the care grade was added to the German social security system in order to ensure appropriate long-term care [19]. With significantly more patients with sarcopenia having a grade of care compared to those without sarcopenia, an influence on disability and an impressive decline in independence was once again confirmed. Also for the first time, a significant association between fewer GR and sarcopenia was observed, while its structured assessment is not yet part of the clinical routine in Germany [15]. 

A significant association between sarcopenia and depression, lower quality of life, and lower self-esteem during hospitalization and at follow-ups was observed. While a high prevalence of depression symptoms and lower quality of life have already been associated with sarcopenia [29,30,31], the present data confirm these observations and shed light on another under-investigated but highly important factor of resilience and stress resistance in advanced age-self-esteem [12]. Given the established associations between lower self-esteem, reduced physical activity, and frailty [32] and the potential of improved self-esteem to mitigate readmissions and prevent functional decline [33,34], it is imperative to acknowledge the interplay between sarcopenia and low self-esteem in sarcopenia management.

In this secondary analysis, the prevalence of sarcopenia based on the EWGSOP2 algorithm applied to older multimorbid hospitalized patients showed a high rate of 63% compared to previously reported prevalence ranging between 9% and 51% [35]. However, as we assessed hospitalized patients, this high prevalence is likely due to the geriatric nature and frailty of the studied population. Overall, it remains difficult to compare the exact prevalence of sarcopenia due to the heterogeneity of the study populations as well as the various assessments, definitions, and cut-offs used. Huemer et al. recently suggested using a higher cut-off for low handgrip strength than the one used in EWGSOP2, a measure that could potentially identify more individuals at risk for sarcopenia [36] by preventing early exclusion through grip strength in the first steps of the algorithm. This adjustment would increase the prevalence of sarcopenia even further than that currently observed, presenting even greater public health relevance and promoting even more concern for this largely unmet medical need [2,37,38,39]. Contrariwise, no significant difference could be found regarding rehospitalization or mortality. This, however, is not consistent with the results of previous studies that showed significant correlations between the degree of muscle wastage and increased number of hospitalizations as well as higher mortality [40]. In examining the disparities between our study’s results and those reported in previous research, it is crucial to consider several factors that may account for these variations. Prior studies demonstrating an association between sarcopenia and mortality and analyzing patient groups similar to the group in this study in terms of setting (inpatients), age (older than 65), and sarcopenia definition (EWGSOP2) stand out for their length of follow-up of at least 3 years and larger sample size [41,42]. As mentioned above, a significant association between sarcopenia and number of falls was shown in this analysis. As low strength and low muscle mass contribute to the impairment of balance [43,44] and are therefore associated with falls, and, at the same time, with falls being one of the main mechanisms relating sarcopenia to mortality, it is possible that the length of the follow-up period was decisive for the inconsistent result in terms of mortality in this analysis. It is therefore possible that studies with larger sample sizes and longer follow-up periods are needed to rule out this inconsistency.

Limitations of this analysis: One limitation was the relatively small sample size of this study population, with 97 patients included. However, the patient population was highly phenotypized [12], and several of the observations confirm what is already reported in the literature. Another limitation is the relatively short follow-up period, i.e., 6 months, which was predetermined as per the original RCT protocol. This duration may not have captured the full spectrum of changes and outcomes associated with sarcopenia, most notably when it comes to long-term implications and mortality. Also, the use of a complete case analysis from an RCT limits the generalizability of the results. The limited availability of longer-term, comprehensive longitudinal studies in the field of sarcopenia underscores the need for further research in order to better understand the trajectory and impact of this condition over extended periods. 

## 5. Conclusions

In conclusion, sarcopenia was shown to be significantly associated with MPI prognosis, PROMs (depression, quality of life, self-esteem, and GR), and adverse outcomes (falls, grade of care), independent of age, gender, and TIDP. This suggests that sarcopenia has prognostic significance for older hospitalized patients, and its consideration might be useful for shared decision making in a clinical setting. With the prevalence of sarcopenia among older adults being high and increasing and with this condition, in fact, being underdiagnosed [27,45], the newly established definition for the diagnostic steps of sarcopenia appears particularly important for systematic early detection and management of this highly unfavorable disorder.

## Figures and Tables

**Figure 1 jcm-13-03116-f001:**
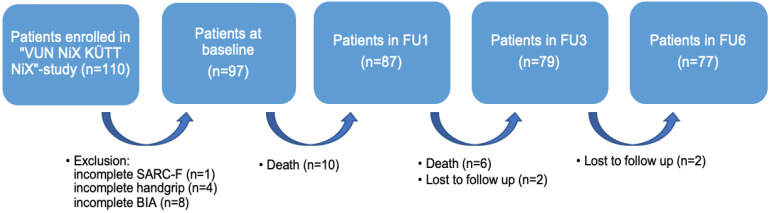
Flow chart for the secondary analysis. Notes: FU = Follow-Up; SARC-F = strength, assistance with walking, rise from a chair, climb stairs, and falls; BIA = Bioelectrical Impedance Analysis.

**Figure 2 jcm-13-03116-f002:**
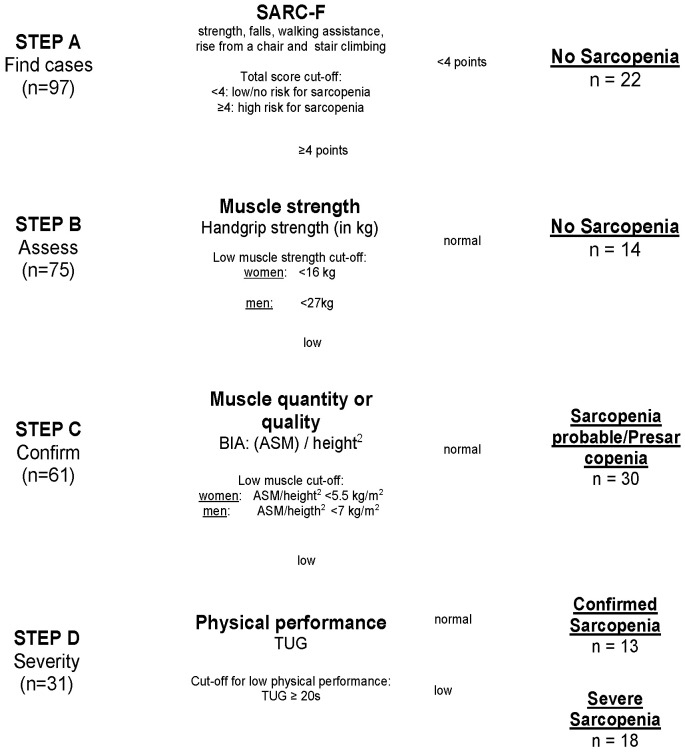
Flowchart of study population after EWGSOP2 stepwise procedure for the diagnosis of sarcopenia. Notes: SARC-F = strength, assistance with walking, rise from a chair, climb stairs, and falls; BIA = Bioelectrical Impedance Analysis; ASM = Appendicular Skeletal Mass; TUG = Timed Up-and-Go test.

**Figure 3 jcm-13-03116-f003:**
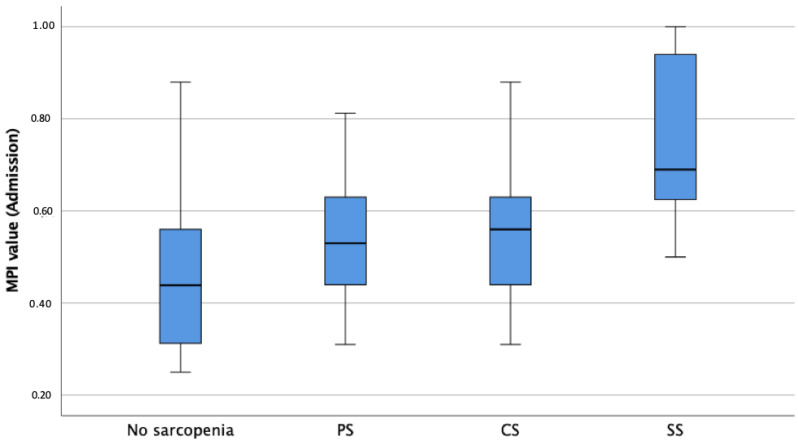
Boxplots showing sarcopenia groups according to EWGSOP2 in association with MPI-prognosis at admission. Notes: MPI = Multidimensional Prognostic Index; PS = Probable Sarcopenia; CS = Confirmed Sarcopenia; SS = Severe Sarcopenia.

**Table 1 jcm-13-03116-t001:** Comparison of sarcopenic and non-sarcopenic patients according to EWGSOP2 criteria for sarcopenia.

		Total (n = 97)	No Sarcopenia (n = 36; 37%)	Sarcopenia (n = 61; 63%)	Probable Sarcopenia (PS) (n = 30; 31%)	Confirmed Sarcopenia (CS) (n = 13; 13%)	Severe Sarcopenia (SS) (n = 18; 19%)	*p*-Value ^a,b^
Age (years), mean (SD)		76.5 (6.8)	76.3 (6.9)	76.7 (6.8)	77.3 (7.2)	75.7 (8.0)	76.6 (5.2)	0.752
Female, n (%)		53 (55)	25 (70)	28 (46)	10 (36)	7 (47)	11 (61)	0.024
Length of hospital stay (days), mean (SD)		20.4 (12.5)	17 (7.2)	22.3 (14.5)	24.2 (18.8)	19.3 (8.1)	22 (10.8)	0.043
Number of GCT (days), median (IQR)		16 (11–22)	14.5 (8–17)	17.0 (15–23)	21.5 (14.8–23)	17.0 (13.5–19)	15.0 (14.5–22.3)	0.006 *
Grade of care, n (%)		65 (75)	19 (56)	46 (89)	20 (83)	12 (86)	14 (93)	<0.001 *
BMI, median (IQR)		24.2 (21.2–27.2)	24.1 (19.9–27.05)	24.4 (21.4–27.26)	26.4 (22.55–30.32)	23.6 (21–27)	21.5 (18.93–26.4)	0.563
Hospitalized in the last year, n (%), n = 95		78 (82)	25 (71)	53 (88)	25 (86)	12 (92)	16 (89)	0.038
Falls in the last year, n (%)		52 (54)	13 (36)	39 (64)	17 (57)	11 (85)	11 (61)	0.008
Scores upon admission								
MPI Value, median (IQR)		0.56 (0.44–0.69)	0.43 (0.31–0.56)	0.56 (0.44–0.69)	0.53 (0.44–0.63)	0.56 (0.44–0.66)	0.69 (0.63–0.94)	<0.001 **
MPI-Group, n (%)	MPI-1	15 (15)	11 (31)	4 (7)	3 (11)	1 (6)	0 (0)	<0.001 **
	MPI-2	56 (58)	21 (58)	35 (57)	18 (64)	10 (67)	7 (39)	
	MPI-3	26 (27)	4 (11)	22 (36)	7 (25)	4 (27)	11 (61)	
ADL, mean (SD)		3.9 (1.8)	5.1 (1.0)	3.2 (1.7)	3.5 (1.8)	3.6 (1.4)	2.6 (1.7)	<0.001 *
IADL, mean (SD)		4.1 (2.6)	5.9 (2.2)	3.1 (2.3)	3.6 (2.3)	2.9 (2.4)	2.4 (2.0)	<0.001 *
CIRS, mean (SD)		6.3 (1.8)	5.7 (1.8)	6.7 (1.8)	6.3 (1.5)	6.2 (1.7)	7.5 (2.0)	0.011 *
MNA-SF, mean (SD)		7.3 (3.0)	8.2 (3.0)	6.8 (2.8)	7.8 (2.7)	6.7 (2.8)	5.2 (2.4)	0.024 *
SPMSQ, mean (SD)		1.64 (2.18)	0.92 (1.13)	2.1 (2.5)	1.47 (1.59)	2.23 (2.68)	3.00 (3.45)	0.011 *
ESS, mean (SD)		15.3 (2.7)	16.8 (2.3)	14.4 (2.7)	15.2 (1.9)	13.5 (2.8)	13.6 (2.6)	0.003 *
Number of medications, mean (SD)		10.7 (4.5)	9.6 (3.6)	11.4 (4.5)	11.9 (5.3)	9.9 (4.0)	11.5 (4.7)	0.224 *
EQ-5D-5L, median (IQR)		0.62 (0.37–0.85)	0.79 (0.57–0.92)	0.53 (0.29–0.76)	0.55 (0.28–0.79)	0.57 (0.33–0.70)	0.52 (0.30–0.64)	<0.001 *
GDS, median (IQR), n = 96		3.5 (2–6)	3 (2–5)	4.5 (2–7)	3 (2–7)	6 (3.5–11.5)	4 (2–5.5)	0.696 *
RSES, mean (SD)		25.2 (3.8)	26.3 (3.0)	24.5 (4.0)	25.3 (3.3)	23.2 (5.2)	24.2 (4.1)	0.025 *
GS, mean (SD)		7.1 (2.2)	6.3 (2.1)	7.6 (2.2)	7.4 (2.1)	7.8 (2.0)	7.7 (2.4)	0.033 *
GR, mean (SD)		7.3 (2.2)	8.2 (1.7)	6.8 (2.3)	7.4 (2.1)	6.1 (2.3)	6.4 (2.6)	0.003 *
More GR than GS, n (%)		78 (80)	34 (94)	44 (72)	22 (79)	10 (67)	12 (67)	0.007 *
Low Handgrip, n (%)		71 (73)	10 (14)	61 (100)	30 (42)	13 (18)	18 (25)	<0.001 *
TUG test ≥ 20 s, n (%)		63 (81)	27 (87)	36 (77)	17 (90)	1 (10)	18 (100)	<0.001 *
Blood levels								
Albumin upon Admission, g/L, mean (SD)		32.2 (6.1)	35.2 (5.5)	30.5 (5.8)	31.0 (5.4)	29.9 (7.4)	29.9 (5.4)	<0.001
Albumin Discharge, g/L, mean (SD)		31.5 (8.7)	33.9 (8.5)	30.5 (8.7)	29.2 (10.9)	32.0 (5.8)	31.3 (6.9)	0.109
Total Protein upon Admission, g/L, mean (SD)		65.0 (9.4)	66.9 (9.1)	65.0 (9.4)	63.9 (8.6)	61.5 (10.4)	65.7 (10.2)	0.134
Total Protein Discharge, g/L, mean (SD)		63.4 (10.8)	63.9 (15.6)	63.1 (7.6)	61.9 (7.3)	64.9 (5.0)	63.7 (9.5)	0.752
Follow-Up								
Falls, n (%)	during hospital stay, n = 97	8 (8)	1 (3)	7 (12)	4 (13)	2 (15)	1 (6)	0.132 *
	FU1, n = 89	19 (21)	2 (6)	17 (32)	10 (37)	2 (17)	5 (33)	0.004 *
	FU3, n = 83	25 (30)	4 (12)	21 (43)	11 (42)	2 (22)	8 (57)	0.002 *
	FU6, n = 84	28 (33)	4 (12)	24 (48)	12 (44)	2 (22)	10 (71)	0.027 *
EQ-5D-5L, median (IQR)	FU1, n = 68	0.65 (0.34–0.87)	0.83 (0.60–0.95)	0.51 (0.31–0.80)	0.58 (0.31–0.85)	0.62 (0.11–0.82)	0.34 (0.24–0.67)	0.013 *
	FU3, n = 54	0.74 (0.38–0.85)	0.86 (0.66–0.93)	0.59 (0.30–0.79)	0.51 (0.29–0.79)	0.65 (0.1–0.77)	0.63 (0.22–0.84)	0.002 *
	FU6, n = 63	0.65 (0.37–0.85)	0.78 (0.45–0.91)	0.56 (0.37–0.82)	0.71 (0.42–0.86)	0.19 (0.01–0.66)	0.58 (0.43–0.68)	0.043 *
Rehospitalization, n (%)	FU1, n = 85	26 (31)	8 (24)	18 (35)	12 (43)	2 (22)	4 (27)	0.070 *
	FU3, n = 85	35 (41)	12 (36)	23 (44)	14 (50)	4 (44)	5 (33)	0.473 *
	FU6, n = 74	43 (50)	15 (46)	28 (54)	17 (61)	4 (44)	7 (47)	0.451 *
Mortality, n (%)	alive after 1 month	87 (90)	34 (94)	53 (87)	26 (87)	12 (92)	15 (83)	0.240 *
	alive after 3 months	79 (83)	32 (91)	47 (78)	24 (80)	10 (83)	13 (72)	0.268 *
	alive after 6 months	79 (83)	32 (91)	47 (78)	24 (80)	10 (83)	13 (72)	0.268 *
GDS, mean (SD)	FU1, n = 61	4.2 (3.1)	3.2 (2.6)	4.9 (3.4)	4.5 (3.2)	5.6 (4.4)	5.3 (2.7)	0.035 *
	FU3, n = 53	3.9 (3.1)	3.2 (2.6)	4.6 (3.2)	4.5 (3.2)	5.6 (4.4)	5.3 (2.7)	0.080 *
	FU6, n = 53	4.7 (3.1)	4.0 (2.9)	5.3 (3.2)	4.1 (2.5)	6.8 (5.0)	3.6 (2.2)	0.121 *
RSES, median (IQR)	FU1, n = 60	27 (24–28.5)	27 (25–29)	26 (24–28)	26 (24–28)	25.5 (18–27.75)	28 (26.5–29)	0.744 *
	FU3, n = 61	28 (25–29)	29 (27.5–30)	27 (24–28)	27 (24–28)	27 (15.75–27.25)	27 (24–29)	0.002 *
	FU6, n = 53	28 (25–29)	28 (25–29)	27.5 (25–29)	28 (27–30)	24.5 (14.25–28.5)	25 (22–28.75)	0.444 *

Notes: Patients were subdivided into sarcopenia groups using the EWGSOP2 algorithm. SD = standard deviation; IQR = Interquartile Range; GCT = Geriatric Complex Treatment; BMI = Body Mass Index; MPI = Multidimensional Prognostic Index; ADL = Katz’s Activities of Daily Living; IADL = Lawton’s Instrumental Activities of Daily Living; CIRS = Cumulative Illness Rating Scale; MNA-SF = Mini Nutritional Assessment Short Form, SPMSQ = Short Portable Mental Status Questionnaire; ESS = Exton Smith Scale; EQ-5D-5L = EuroQol Quality of Life—5 Dimensions (5-Level Version); GDS= Geriatric Depression Scale; RSES = Rosenberg Self-Esteem Scale; GSs = Geriatric Syndromes; GR = Geriatric Resources; FU = Follow-Up. ^a^ After testing for normal distribution (Kolmogorov–Smirnov) and interpreting the histogram, the Mann–Whitney-U test was performed for medians, *t*-test was conducted for means, and Chi-squared test was conducted for frequencies. ^b^ Indicated *p*-values are those for the comparison between No Sarcopenia and Sarcopenia groups, significant at 5%, significant *p*-values highlighted in bold. * After linear/logistic regression analysis was conducted, results were adjusted for age, sex, and MPI. ** After linear/logistic regression analysis was conducted, results were adjusted for age and sex.

**Table 2 jcm-13-03116-t002:** Post hoc tests for pairwise comparison for MPI subdomains.

		M_Diff_	95%-CI	*p*-Value
ADL				
No Sarcopenia	PS	1.72	0.73–2.71	<0.001 *
	CS	1.47	0.17–2.76	0.018 *
	SS	2.47	1.32–3.63	<0.001 *
PS	CS	−0.25	−1.58–1.08	1.0
	SS	0.76	−0.44–1.95	0.547
CS	SS	1.0	−0.45–2.46	0.398
IADL				
No Sarcopenia	PS	2.35	0.86–3.84	<0.001 *
	CS	3.07	1.12–5.02	<0.001 *
	SS	3.53	1.79–5.27	<0.001 *
PS	CS	0.721	−1.28–2.72	1.0
	SS	1.18	−0.62–2.98	0.667
CS	SS	0.457	−1.17–2.65	0.815
CIRS				
No Sarcopenia	PS	−0.64	−1.79–0.51	0.826
	CS	−0.54	−2.04–0.97	0.559
	SS	−1.81	−3.15–0.46	0.003 *
PS	CS	0.10	−1.44–1.65	1.0
	SS	−1.17	−2.55–0.22	0.154
CS	SS	−1.27	−2.96–0.42	0.277
MNA-SF				
No Sarcopenia	PS	0.361	−1.48–2.20	1.0
	CS	1.502	−0.91–3.91	0.576
	SS	3.03	0.88–5.18	0.002 *
PS	CS	1.14	−1.33–3.61	1.00
	SS	2.67	0.45–4.89	0.010 *
CS	SS	1.53	−1.18–4.24	0.794
SPMSQ				
No Sarcopenia	PS	−0.55	−1.93–0.83	1.00
	CS	−1.31	−3.12–0.49	0.317
	SS	−2.08	−3.73–0.44	0.006 *
PS	CS	−0.76	−2.62–1.09	1.00
	SS	−1.53	−3.23–0.16	0.100
CS	SS	−0.769	−2.83–1.29	1.00
ESS				
No Sarcopenia	PS	1.54	0.00–3.09	0.051
	CS	3.24	1.21–5.26	<0.001 *
	SS	3.17	1.36–4.97	<0.001 *
PS	CS	1.70	−0.38–3.77	0.182
	SS	1.62	−0.24–3.49	0.127
CS	SS	−0.73	−2.35–2.21	0.182

Notes: Bonferroni-corrected post hoc test for parametric values. * *p*-values of pairwise comparison between the groups, significant at level 0.05. MPI = Multidimensional Prognostic Index; M_Diff_ = Mean Difference; CI = Confidence Interval; PS = Probable Sarcopenia; CS = Confirmed Sarcopenia; SS = Severe Sarcopenia; ADL = Katz’s Activities of Daily Living; IADL = Lawton’s Instrumental Activities of Daily Living; CIRS = Cumulative Illness Rating Scale; MNA-SF = Mini Nutritional Assessment Short Form; SPMSQ = Short Portable Mental Status Questionnaire; ESS = Exton Smith Scale.

## Data Availability

The data presented in this study are available on request from the corresponding author. The data are not publicly available due to privacy.

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
