# Peer review of "Prognostic Features of Sarcopenia in Older Hospitalized Patients: A 6-Month Follow-Up Study"

_jcm, 2024, doi:10.3390/jcm13113116_

Round 1

Reviewer 1 Report

Comments and Suggestions for Authors

The article by Anne Ferring and colleagues provides a 6-month follow-up study that assesses the prevalence and prognostic factors of sarcopenia in older hospitalized patients. The authors addressed the main question of how prevalent sarcopenia is in hospitalized older adults and how it is linked to their prognosis and overall well-being. 

As sarcopenia is associated with adverse health outcomes, this study focuses on hospitalized patients, a population potentially more susceptible to illness and reduced mobility. This helps the understanding of sarcopenia in a specific clinical setting. This 6-month follow-up study used data from 97 (out of 110) hospitalized patients aged 60 and older who were diagnosed with multiple chronic conditions. Sarcopenia was diagnosed using the European Working Group on Sarcopenia in Older People (EWGSOP2) criteria. The authors also assessed patients' multidimensional frailty and prognosis using the Comprehensive Geriatric Assessment (CGA). This study further adds to the field by demonstrating a strong association between sarcopenia and negative outcomes in hospitalized older adults. It highlights the importance of considering sarcopenia in this population for better patient care.

The conclusions are consistent with the evidence presented in Tables 1 and 2. The high prevalence of sarcopenia and its association with poorer prognosis, mood, and quality of life are well-supported by the data on frailty scores, geriatric syndromes, and patient resources.

One methodological improvement could be a larger sample size (more than 97) and a more diverse patient population. Additionally, a longer follow-up period beyond 6 months would provide a more comprehensive picture of long-term outcomes.

The language is clear and professional throughout the article. 

The reference list is comprehensive and includes relevant literature. The authors need to include the most recent publications (2023-2024) in this study and discuss the results and limitations in the “Discussion” section.

In Figure 2, the authors should increase the font size for better readability.

Reviewer 2 Report

Comments and Suggestions for Authors

22 April 2024

Dear Editor of JCM

Thank you for your invitation me to review the manuscript entitled “Prognostic features of sarcopenia in older hospitalized patients: a 6-month follow-up study”. This is a well-written manuscript. I appreciate the steps in the method part that are appropriately presented. However, I have one question. Why authors stated that this study was a RCT, if so, what did the control group receive? Authors divided subjects into sarcopenia and non-sarcopenia, so this should not be the RCT design. Please consider.

Best Regards,

Reviewer

Reviewer 3 Report

Comments and Suggestions for Authors

The research article titled <<Prognostic features of sarcopenia in older hospitalized 2 patients: a 6-month follow-up study>> is well written. 

However, there are minor issues. 

1) Please create an image to describe the study design or transfer Figure 1 in the methods area. 

2) Please present a chart with handgrip strength results

3) Please present a chart with TUG test results

Author Response

Please see the attchment. 
